# Prevalence, Antibiotic Resistance, Toxin-Typing and Genotyping of *Clostridium perfringens* in Raw Beef Meats Obtained from Qazvin City, Iran

**DOI:** 10.3390/antibiotics11030340

**Published:** 2022-03-04

**Authors:** Samaneh Hassani, Babak Pakbin, Wolfram Manuel Brück, Razzagh Mahmoudi, Shaghayegh Mousavi

**Affiliations:** 1Medical Microbiology Research Center, Qazvin University of Medical Sciences, Qazvin 34197-59811, Iran; samanehasani1369@gmail.com (S.H.); sh.mousavi@qums.ac.ir (S.M.); 2Institute for Life Technologies, University of Applied Sciences Western Switzerland Valais-Wallis, 1950 Sion, Switzerland; wolfram.bruck@hevs.ch

**Keywords:** *Clostridium perfringens*, raw meat, antibiotic resistance, toxin gene

## Abstract

Background: *Clostridium perfringens* is one of the highest prevailing spore-forming foodborne pathogens, which is widely distributed and causes severe disease and outbreaks in humans and animals. Raw meat and poultry are the main vehicles of this pathogen. In this study, we investigated the prevalence, antibiotic resistance pattern, toxin-encoding genes and genetic diversity of *C. perfringens* isolates from raw whole and minced meat samples purchased from local markets in Qazvin city, Iran (the source of beef cattle production was also located in Qazvin city, Iran). Methods: We used conventional culture-based and Kirby–Bauer disk diffusion and conventional and arbitrary primer PCR methods. Results: A total of 18 *C. perfringens* strains were isolated from 133 raw meat samples (13.53%). Up to 44.4 and 55.5% of these isolates were detected in raw minced and whole meat samples, respectively. We found that 72.2, 66.6, 61.1, 37.8 and 33.3% of the *C. perfringens* isolates were resistant to ampicillin, tetracycline, amoxicillin, ciprofloxacin and chloramphenicol antibiotics, respectively. Multidrug resistance was found in 38% of the isolates. Among the four main toxin genes evaluated, the *Cpa* gene was detected in all isolates, and 61.1% of the isolates were mostly recognized as type A *C. perfringens*. High levels of genetic diversity were observed among the isolates, and they were classified into five distinct groups. Conclusions: The isolates from whole meat samples were more resistant to antibiotics. However, toxin genes were more detected in the isolates from minced meat samples. Our findings suggest that contamination of raw meat products with multidrug resistant *C. perfringens* could be regarded as one of the concerning pathogens in these products. Comprehensive monitoring of *C. perfringens* isolates is strongly recommended.

## 1. Introduction

*Clostridium perfringens* is a ubiquitous rod-shaped, Gram-positive, nonmotile anaerobe that grows rapidly and is one of the highest prevailing spore-forming pathogenic bacteria with a worldwide distribution [1]. It has been found in various environments and is present in foods (raw or cooked under anaerobic conditions), sewage, dust and soil. Raw meat and poultry are recognized as the main vehicles of the foodborne diseases caused by *C. perfringens* [2]. It is estimated that *C. perfringens*, known as one of the most prevalent bacterial pathogens, causes more than one million foodborne illnesses in the United States [1]. However, this pathogen caused several foodborne disease outbreaks in Japan, England, Australia and Wales. These outbreaks were caused mainly by *C. perfringens* strains and were frequently associated with consuming undercooked or raw contaminated meat and poultry products [3]. 

*C. perfringens* causes toxico-infectious diseases, such as gastroenteritis and acute diarrhea, in humans, in which their toxins play an important role [4]. This pathogen can produce and release up to 16 various toxins in different combinations. However, four major toxins, namely alpha, beta, epsilon and iota, which are encoded by the *cpa*, *cpb*, *etx* and *iot* genes, respectively, are produced and secreted by *C. perfringens* isolates, causing intestinal diseases [5]. According to the main toxin profile produced by *C. perfringens*, this pathogen is classified into five distinct toxinotypes consisting of A, B, C, D and E. Most diseases caused by *C. perfringens* strains are mediated by the combination of one or more of these toxins [6]. The alpha toxin is essential for cases of gas gangrene in diseased humans and animals. The *Cpb* gene encodes a toxin responsible for enterotoxemia and necrotizing enteritis, predominantly in the neonates of some animals. The epsilon toxin is responsible for lesions and clinical signs of enterotoxemia, a common neurological disease of goat and sheep caused by *C. perfringens*. The iota toxin mediates the pathogenesis of the intestinal diseases in humans caused by type E *C. perfringens* [4]. Regarding the fact that the specific toxinotypes of *C. perfringens* strains are associated with specific intestinal and extraintestinal diseases, toxinotyping these pathogens is extremely important [7].

Antibiotics are currently the main treatment of bacterial infectious diseases in humans and animals to decrease the mortality and morbidity associated with disease. Therefore, antibiotic resistance in foodborne bacterial pathogens, and especially those isolated from animal-based foods, have gradually increased all over the world [8]. Irrational and excessive use of various classes of antibiotics for the treatment of infections and the promotion of growth of livestock and farm animals are the main causative factors leading to an increase in antibiotic resistance among foodborne pathogens [9]. Antibiotics such as metronidazole, chloramphenicol, ampicillin, imipenem and tetracycline have been used to decrease the economic losses caused by infectious diseases in the livestock industry and farm animals [10]. Several studies reported that *C. perfringens* isolated from food samples were mostly resistant to tetracycline, erythromycin and lincomycin antibiotics over the recent decades [11]. When bacterial pathogens are resistant to more than three classes of antibiotics, they are known as multidrug resistant (MDR). Nowadays, one of the major concerns in food safety and public health is the emergence of MDR foodborne pathogens [12]. 

Genotyping methods have been used to determine the genetic relatedness and diversity among the foodborne pathogens, especially in outbreaks [13]. Several common genotyping assays have been used to type *C. perfringens* isolates from food samples, including multiple-locus variable-number tandem repeat analysis (MLVA), multi-locus sequence typing (MLST), amplified polymorphic DNA (RAPD), pulsed-field gel electrophoresis (PFGE), amplified fragment length polymorphism (AFLP), arbitrary primer PCR-based (AP-PCR) and toxinotyping methods [14,15]. There are limited studies to characterize the antibiotic resistance, presence of toxin genes and genetic relatedness in *C. perfringens* isolated from raw meat samples [16,17,18,19]. Therefore, the purpose of this study was to determine the prevalence rate, antimicrobial susceptibility, toxinotype and genetic diversity in *C. perfringens* isolates from raw whole and minced meat samples.

## 2. Results

### 2.1. Isolation and Identification of C. perfringens in Raw Meat Samples

In this study, *C. perfringens* were isolated from a total of 18 out of 133 (13.53%) raw meat samples. The prevalence rate of *C. perfringens* isolated from minced and whole meat samples is shown in Figure 1. All *C. perfringens* isolates were initially isolated using culture-based methods and then confirmed and identified using biochemical tests. Among all *C. perfringens* isolates, 8 out of 18 (44.44%) and 10 out of 18 (55.55%) isolates were detected in raw minced and whole meat samples, respectively. The prevalence rate of *C. perfringens* was significantly (*p* < 0.05, chi-square test) higher in raw whole meat than that in the raw minced meat samples.

### 2.2. Antimicrobial Susceptibility Testing of the C. perfringens Isolates

All eighteen *C. perfringens* isolates were tested for antibiotic resistance against six diverse classes of antibiotics and nine different commercial antibiotics. The results of the antimicrobial susceptibility testing of the isolates are illustrated in Table 1. In total, 13 (72.2%), 12 (66.6%), 11 (61.1%), 7 (37.8%) and 6 (33.3%) out of the 18 *C. perfringens* isolates from raw meat samples were resistant to ampicillin, tetracycline, amoxicillin, ciprofloxacin and chloramphenicol antibiotics, respectively. The lowest levels of antibiotic resistance were seen against imipenem (5 out of 18; 27.7%), ceftriaxone (4 out of 18; 22.2%), amikacin (3 out of 18; 16.6%) and cefepime (3 out of 18; 16.6%) in the *C. perfringens* isolates from all raw meat samples. Notably, significantly (*p* < 0.05) higher levels of antibiotic resistance were observed in the *C. perfringens* isolates from the raw whole meat samples than that in the raw minced meat samples. In this study, we found that 7 out of 18 (38.8%) *C. perfringens* isolates from all raw meat samples were resistant to at least three different classes of antibiotics and were considered to be MDR *C. perfringens* isolates (Table 2). The frequency of multidrug resistance patterns among the *C. perfringens* isolates were similar (n = 1). The group of isolates that was resistant to three classes of antibiotics was the most frequent one (n = 3). Notably, diverse patterns of resistance to different antibiotic classes were observed among the *C. perfringens* isolates (Table 2).

### 2.3. Toxin-Encoding Genes in C. perfringens Isolates

Toxin-encoding genes, including the *cpa*, *cpb*, *cpe*, *etx* and *iap* genes, were detected and identified in *C. perfringens* isolated from the raw whole and minced meat samples by a conventional multiplex PCR assay using specific primers. All *C. perfringens* isolates (18 out of 18; 100%) harbored the *cpa* gene encoding the alpha toxin. Five *C. perfringens* isolates (27.7%) harbored *cpe* or *etx*, two isolates (11.1%) harbored *cpb* and only one isolate (5.5%) harbored the *iap* toxin-encoding genes (Table 3). Toxin genes were significantly (*p* < 0.05) more detected in the *C. perfringens* isolates from the raw minced meat samples than those from the raw whole meat samples. Moreover, 61.1 and 22.2% (11 out of 18) of *C. perfringens* isolates were identified as toxinotypes A and D, respectively, in this study (Table 3).

### 2.4. Genotyping and Molecular Toxinotyping of the C. perfringens Isolates

Genotyping of the *C. perfringens* isolates from the raw meat samples was performed by conventional PCR using arbitrary OPA-3 primers. The method was able to type all isolates (18 out of 18; 100%), which were differentiated into five distinct groups (O1, O2, O3, O4 and O5), indicating the genetic diversity that exists among the different *C. perfringens* isolates (Figure 2). According to Simpson’s index of diversity, the discriminating capability of the PCR method with the OPA-3 primer for the genotyping of the *C. perfringens* isolates in this study was relatively high (73.2%) regarding the 50% similarity coefficient. The genetic relatedness among the isolates ranged from 50 to 74%, indicating a high level of genetic variation (Figure 2). Any significant differences in banding profiles were considered to differentiate between two OPA-3 typing groups. 

Based on the source of the isolates, the resistance phenotype, presence of toxin genes, toxinotypes and genotypes in the isolates are shown in Table 3. The isolates from the raw minced meat samples were included in all genotyping clusters (O1–O5). The isolates from the whole meat samples were only detected in the groups O1 and O2, indicating that there are higher levels of genetic diversity among the *C. perfringens* isolates from the minced meat samples. Clusters O4 and O5 contained one isolate each. The greatest number of isolates (8 out of 18; 44.4%) were included in cluster O1. Three types of toxins, including the A, Ae and D toxinotypes, were detected in the isolates from the whole meat samples. Five different types of toxins consisting of the Ae, B, Ce, D and E toxinotypes were seen in the *C. perfringens* isolates from the minced meat samples. The A and E toxinotypes were only detected in the isolates from the raw whole and minced meat samples, respectively. Moreover, higher levels of toxinotype diversity and variation in the presence of different toxin genes were seen among the isolates from the raw minced meat samples. In addition, higher levels of antibiotic resistance were observed in the *C. perfringens* isolates from the raw whole meat samples (Table 3).

## 3. Discussion

*C. perfringens* is the main cause of human gas gangrene and some major important foodborne diseases in humans [20]. This pathogen also has the ability to form spores protecting the bacterial cell against stress conditions, such as exposure to heat and oxygen, allowing *C. perfringens* to survive in varied environments and through thermal processes, including sterilization and cooking, allowing it to reach the high levels that are needed to cause food poisoning [21]. Due to these properties, *C. perfringens* has been regarded as an important foodborne pathogen. *C. perfringens* is widely distributed in the gastrointestinal tracts of humans, animals and soil as a ubiquitous pathogen [4]. *C. perfringens* endospores can be transmitted via food, water and food commodities to humans, causing foodborne diseases. Foods, especially raw meat products, are the main transmission vehicle of this pathogen to humans [22]. Poor hygiene and sanitation and insufficient thermal processing conditions during food production and distribution contribute to the increase in the prevalence rates of *C. perfringens* in meat products and also lead to an increase in the incidence of foodborne diseases caused by this pathogen [23]. Limited studies investigated the antimicrobial susceptibility and toxinotype profiles in *C. perfringens* isolated from food samples. Therefore, we conducted this research to determine the toxinotypes, antibiotic resistance and genotypic relationship among these properties in *C. perfringens* isolated from raw meat samples, as they are the main foods contaminated with this foodborne pathogen. 

Few studies are currently available regarding the prevalence of *C. perfringens* in foods as well as raw meat products [24]. In this study, the total prevalence rate of *C. perfringens* was significantly higher than that reported from Côte d’Ivoire (12.4%; 49 out of 395 food samples) [25], Kazakhstan (9%; 18 out of 197 food samples) [26] and Nigeria (13.18%; 29 out of 220 food samples) [16] and lower than that reported from Argentina (24.46%; 126 out of 515 food meat samples) [27], Japan (71.0%; 143 out of 200 total different meat products) [28], Turkey (92%; 92 out of 100 ground beef and sheep meat samples) [29], China (23.1 and 15.1%; 130 and 38 out of 562 broiler chicken and 252 retail chicken samples, respectively) [30] and Korea (19%; 38 out of 200 chicken, beef and pork meat samples) [31]. These differences may be because of strong variations in hygienic and sanitary conditions of handling, processing and distribution of the products [32]. However, the results of the *C. perfringens* prevalence rate in the raw meat samples showed that, regarding the lower prevalence rate, there is still a high risk with respect to contamination with this pathogen in raw meat products due to poor hygiene and low levels of sanitation practices that have been used during raw meat processing and distribution [19,31]. The results in this study also showed that the prevalence rate of *C. perfringens* was significantly (*p* < 0.05, chi-square test) higher in raw whole meat samples than that in minced meat samples. Notably, depending on the type of meat used, minced meat is more likely to be contaminated than whole meats due to the extra handling throughout the grinding process and the release of meat juice that allows spoilage bacteria and foodborne pathogens to multiply [33]. However, only a very limited number of studies investigated differences in the prevalence of *C. perfringens* between minced and whole meat samples. Previously, in Argentina, Stagnitta et al. [27] reported that minced meat was more contaminated with *C. perfringens* than other raw meat products. In another study that was conducted recently by Yibar et al. [34], *C. perfringens* was isolated in higher levels in raw minced meat and beef meatball than that in other raw meat products. The results obtained in this study suggested that higher prevalence rate of *C. perfringens* may not necessarily imply substantial *C. perfringens* contamination in raw meat products in Iran. Instead, using accurate and efficient isolation and identification assays might have contributed to a significantly higher recovery of *C. pefringens* from raw meat samples.

Antibiotic resistance genes could be transmitted horizontally between various foodborne bacterial pathogens via conjugative plasmids [35]. These genes also could be transferred from various environments and animal-based foods to the commensal microbiota and opportunistic pathogens in human and animals through food chains [36]. There were also few studies on the antimicrobial susceptibility evaluation of *C. perfringens* isolates from food sources [37]. A study in Korea showed high levels of antibiotic resistance in *C. perfringens* isolated from pork, duck, chicken and beef meat samples and they found that 100, 97, 97, 93, 83, 73, 20, 10, 7, 7 and 3% of the isolates were resistant against ampicillin, penicillin, bacitracin, tetracycline, erythromycin, oxytetracycline, gentamicin, trimethoprim, amikacin and streptomycin antibiotics, respectively [15]. Another study, which was performed in Korea, reported *C. perfringens* strains isolated from pork, chicken and beef meat samples were resistant to tetracycline (38 out of 38 isolates; 100%), imipenem (27 out of 38 isolates; 71%), chloramphenicol (26 out of 38 isolates; 68.4%) and metronidazole (13 out of 38 isolates; 34.2%) [31]. Another study, which was conducted recently by Anju et al. [38], reported that 44, 40, 40 and 26.6% of *C. perfringens* isolated from livestock and poultries were resistant to gentamicin, erythromycin, bacitracin and tetracycline antibiotics, respectively. Several previous studies showed that *C. perfringens* isolated from food samples were mostly susceptible to ciprofloxacin and chloramphenicol, which corresponds to our findings. Higher resistance to tetracycline of *C. perfringens* isolates that were reported in other studies is due to the excessive use of this antibiotic and incorrect veterinary advice [39]. We also found that *C. perfringens* isolates from raw whole meat samples were more resistant than isolates from minced meat samples. Recently, resistance to various classes of antibiotics have been developed worldwide in *C. perfringens* isolated from food and clinical samples [24,38]. The prevalence rate of MDR *C. perfringens* isolated from raw meat samples was significantly lower than that of MDR isolates from raw animal-based food samples reported in Korea (78.9%) [31] and China (90.1%) [30]. Lower antibiotic resistance and prevalence rates of MDR *C. perfringens* isolated from raw meat samples is probably due to the rational, effective and appropriate usage of antibiotics in livestock production and treatment of diseases animals in farms [11,37,38]. 

*C. perfringens* is categorized into five distinct toxinotypes (A–E) based on the production of four major toxins, including the alpha, beta, epsilon and iota toxins, which are encoded by the *cpa*, *cpe*, *etx* and *iot* genes, respectively [5,23,31]. However, different strains of this pathogen can produce and release up to 16 various toxins in different combinations, such as the perfringolysin O and beta2 toxins [6]. There are also limited studies on detection of the four main toxin-encoding genes and the determination of toxinotypes in *C. perfringens* isolates collected from food samples. In Turkey, Erol et al. [40] reported that all *C. perfringens* isolates from turkey meat samples carried the *cpa gene*, all of them were recognized as toxinotype A, and none of the other toxin genes were detected. A few years later, Guran and Oksuztepe [41] detected the *cpe* and *cpb2* toxin genes in *C. perfringens* isolates from turkey meat samples for the first time in Turkey. Another study, which was performed in Iran by Afshari et al. [42], reported 70.9 (22 out of 31) and 29% (9 out of 31) of *C. perfringens* isolates from broiler meat samples as types C and A, respectively. A study that has recently been conducted in China by Zhang et al. [30] showed type A as the most predominant toxinotype in 168 *C. perfringens* isolates from chicken meat samples. Another study, which was recently implemented in Korea by Jang et al. [31], revealed that the *cpa* gene was predominantly detected in all 38 *C. perfringens* isolates collected from retail meat samples. Consequently, all isolates were recognized as type A. Our findings correspond with the previous studies performed in Turkey [29], Korea [31] and China [30] and are in disagreement with the results of the study that was previously conducted in Iran [42]. Detection of the *cpa* gene in almost all *C. perfringens* isolates from different types of raw meat samples indicates that it might be a specific universal gene in these isolates [43]. This is the first of type E *C. perfringens* in raw meat samples, as none of the other studies have isolated such a strain. It is worthwhile to note that the *cpa* gene is located on the chromosome. Other toxin genes, including *cpb*, *etx* and *iot,* are plasmid borne, except for the *cpe* gene, which may be located on either plasmids or the chromosome [6]. The acquisition or loss of these plasmids must explain the toxinotype changes seen in the isolates [39]. The *C. perfringens* isolates in this study might have lost their plasmids, as mobile genetic elements containing the toxin genes except *cpa,* which explains why the other toxin genes were not detected in the isolates. The genes encoding the toxins of *C. perfringens* are of great importance from a global public health perspective and are associated with gastrointestinal disorders in humans, such as watery and acute diarrhea, abdominal cramping and necrotizing enteritis [4,5].

To determine the genetic diversity and clonal relatedness among the pathogenic bacterial isolates from food sources, several DNA fingerprinting methods based on arbitrary primers methods have been widely used [44]. It has been shown that these methods are highly efficient to discriminate different genotypes and identify genetic clusters of pathogens that are associated with foodborne disease outbreaks [45]. In the present study, we used a conventional PCR method using an arbitrary OPA-3 primer for the genotyping of *C. perfringens* isolates collected from raw whole and minced meat samples. In this study, we differentiated the *C. perfringens* isolates into five distinct groups (O1–O5) and observed an appropriate discriminatory index of 0.73. In contrast with our findings, Chukwu et al. [19] found 44.7% of the *C. perfringens* isolates were typeable using PCR-based genotyping with arbitrary OPA-3 primers. Llanco et al. [46] found all (100%) of the *C. perfringens* isolates typeable, which is in agreement with our findings. Due to fact the raw meat samples were non-outbreak related and that random and different types of samples were analyzed [47], a relatively wide genetic diversity among the *C. perfringens* isolates was expected in this study. However, the results of the genotyping in our study could not precisely establish a definite and significant relationship between the origin and source of the isolates in the dendrogram [19]. However, we found a relationship between the antibiotic resistance, toxinotype, OPA-3 genotype profiles and the type of raw meat sample (whole and minced) among the *C. perfringens* isolates in this study. We also found that antibiotic resistance, the presence of toxin genes and the genotyping patterns between the isolates from whole and minced meat samples were significantly different, and the cause of this difference was not clear and has not been investigated yet. However, we believe that the main limitation of this study is the small sample size. Consequently, more studies are highly recommended to investigate the reasons for the differences between the molecular and phenotypic characteristics of *C. perfringens* isolated from raw whole and minced meat samples.

## 4. Materials and Methods

### 4.1. Sample Collection

A total of 133 raw meat samples, including 81 minced and 52 whole meat samples, taken from the shank and tenderloin parts of slaughtered beef cattle, were purchased and collected from 46 different local markets located in several areas throughout Qazvin city, Iran, between March and July 2019. All raw meat samples were collected in separated and UV sterilized plastic bags and containers and transported immediately in cool boxes containing ice packs to the laboratory of food microbiology of Qazvin University of Medical Science for further microbiological analysis.

### 4.2. C. perfringens Isolation

*C. perfringens* were isolated and identified in raw minced and whole meat samples according to the method that has been previously described by Chukwu et al. [16]. All raw meat samples were inoculated into cooked meat broth (HiMedia, Mumbai, India) and incubated anaerobically at 37 °C for 24 h for enrichment. Next, 100 µL of bacterial growth was aliquoted to the plates containing tryptose sulphite cycloserine agar (TSC, HiMedia, Mumbai, India), Columbia blood agar (CBA, HiMedia, Mumbai, India) and *Clostridium perfringens* agar (CPA, HiMedia, Mumbai, India) and incubated at 37 °C for 48 h in an anaerobic atmosphere generated using Gas Pack A (Merck, Darmstadt, Germany) in an anaerobic jar (MahAzma, Terhan, Iran) according to the manufacturer’s instructions. Presumptive colonies of *C. perfringens* including black colonies on CPA or TSC and the typical colonies with a double zone of beta-hemolysis on CBA were selected for further biochemical and morphological identification. All confirmed *C. perfringens* isolates were stocked in bovine heart infusion broth (BHI, HiMedia, Mumbai, India) containing 20% (*v*/*v*) glycerol, incubated at 37 °C in an anaerobic atmosphere for 24 h and stored at −80 °C for subsequent analysis. *C. perfringens* ATCC 13124 was used as the positive control in this study. This control strain was activated in BHI broth (HiMedia, Mumbai, India) and incubated anaerobically at 37 °C for 24 h.

### 4.3. Antimicrobial Susceptibility Testing

The antimicrobial susceptibility of the *C. perfringens* isolates was evaluated using the Kirby–Bauer disk diffusion method based on interpretive criteria and the standards established and developed previously by the Clinical and Laboratory Standards Institute [17]. In this study, nine commercial antibiotic disks (Oxoid, Basingstoke, UK) were used, including 30 µg of cefepime (FEP), 10 µg of ampicillin (AMP),25 µg of amoxicillin (AMX), 10 µg of imipenem (IPM), 30 µg of amikacin (AMK), 30 µg of chloramphenicol (CHL), 30 µg of tetracycline (TET), 30 µg of ceftriaxone and 5 µg of ciprofloxacin. The results of the antibiotic susceptibility testing were recorded and described according to the CLSI standards [18]. The Escherichia coli ATCC 25922, Klebsiella pneumoniae ATCC 700603 and Staphylococcus aureus ATCC 25923 strains were used as the positive and negative controls in this study [48].

### 4.4. DNA Extraction

All isolates were grown anaerobically in BHI broth (HiMedia, Mumbai, India) overnight at 37 °C. Next, 1 mL of the bacterial suspension was mixed with the same volume of phosphate buffered saline (PBS, HiMedia, Mumbai, India) and centrifuged at 6000× *g* for 5 min. After the removal of supernatant, the microbial sediment was subjected to DNA extraction using a SinaClon bacterial Gram-positive DNA extraction kit (SinaClon Co., Tehran, Iran) according to the manufacturer’s instruction. The quality and quantity of the extracted genomes were evaluated spectrophotometrically using a NanoDrop model ND-1000 (ThermoFisher Scientific, Waltham, MA, USA). Prior to the PCR reactions, the concentrations of all extracted DNA were adjusted to 50 μg/mL with PBS.

### 4.5. Identification of Toxin Genes

Toxin-encoding genes, including the *cpa*, *cpe*, *cpb*, *etx* and *iap* genes, in the *C. perfringens* isolates were detected and identified using a conventional multiplex PCR assay. Specific primers, which have previously been described by Chukwu et al. [16], were used in this study (Table 4). A 20 µL PCR reaction mixture contained 10 µL of the PCR Master Mix kit (Ampliqon, Odense, Denmark), 0.5 µL of each primer (1 µM/µL), 1 µL of the DNA template (50 µg/mL) and sterilized nuclease free water up to the final reaction volume. The PCR reaction was performed using a Biorad T-100 thermocycler (Biorad, Hercules, CA, USA) that was programmed to: initial denaturation step at 95 °C for 5 min, followed by 30 cycles comprising 95 °C for 45 s, 56 °C for 45 s and 72 °C for 2 min, and a final extension step at 72 °C for 6 min. The PCR products were characterized using electrophoresis in 1.5% *w*/*v* agarose gel containing DNA safe stain (Invitrogen, Carlsbad, CA, USA) at 100 V for 45 min and photographed using the Novin-Pars Gel Documentation system (NovinPars Co., Tehran, Iran). The *C. perfringens* ATCC 13124 strain (*cpa* gene positive) was used as the control.

### 4.6. Determination of Genetic Diversity

The genetic diversity of the *C. perfringens* isolates was measured by the PCR method using the arbitrary primer OPA-3 (Table 4), which has previously been described by Chukwu et al. [19]. Amplifications were performed in 25 µL reaction volumes containing 10 µL of the PCR Master Mix kit (Ampliqon, Odense, Denmark), 1 µL of the primer (5 µM/µL), 2 µL of the DNA template (50 µg/mL) and deionized nuclease free water up to the final reaction volume. The PCR was performed as follows: initial denaturation cycle at 95 °C for 5 min; 35 cycles of 95 °C for 5 min, 42 °C for 1 min and 72 °C for 2 min; and a final extension step at 72 °C for 10 min. The amplified products were characterized using 1.2% *w*/*v* agarose gel electrophoresis at 80 V for 2 h. The gels were stained with DNA safe stain (Invitrogen, Carlsbad, CA, USA). The gels were visualized, and the OPA-3 patterns were recorded by an electrophoresis gel documentation system (NovinPars Co., Tehran, Iran). The OPA-3 markers were analyzed using PyElph software version 1.4 [20]. The dendrogram was generated based on the UPGMA clustering (Dice coefficient) of the OPA-3 profiles using NTSYS-pc software version 2.1 [21]. The OPA-3 patterns of the *C. perfringens* isolates with a similarity index higher than 0.5 were considered to be closely related OPA-3 pattern groups. Simpson’s index of genetic diversity was used to evaluate the discriminating power of the OPA-3 PCR assay in genotyping local isolates of *C. perfringens* as described by Chukwu et al. [19].

### 4.7. Statistical Analysis

A chi-square test was used to measure the significant differences (*p* < 0.05) between the prevalence rates by SPSS software version 21.0.1 (IBM Corp., Armonk, NY, USA). All experiments and measurements were carried out in triplicate.

## 5. Conclusions

In conclusion, we determined the prevalence rate, antimicrobial susceptibility, toxin type and genetic relatedness of *C. perfringens* isolates from raw whole and minced beef meat samples collected from local stores in Qazvin city, Iran from March to July in 2019. In this study, we isolated *C. perfringens* from raw meat samples. The isolates showed a high level of resistance to ampicillin, tetracycline, amoxicillin, ciprofloxacin and chloramphenicol antibiotics. The *cpa* gene, encoding alpha toxin, was identified in all isolates. High levels of clonal diversity were observed among the isolates. Moreover, higher levels of antibiotic resistance were observed among the isolates from raw whole meat samples. Toxin genes were more detected in the isolates from raw minced meat samples. Notably, the sample size in this study was comparably small. Therefore, implementing frequent and comprehensive monitoring, molecular characterization and antimicrobial resistance testing of *C. perfringens* isolates collected from raw meat samples is highly recommended.

## Figures and Tables

**Figure 1 antibiotics-11-00340-f001:**
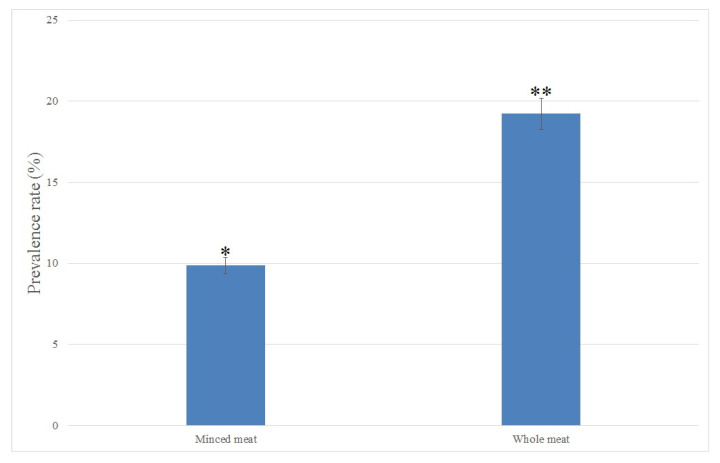
Prevalence rates of *C. perfringens* in the different raw meat samples. * and ** indicate significant differences (*p* < 0.05, chi-square test).

**Figure 2 antibiotics-11-00340-f002:**
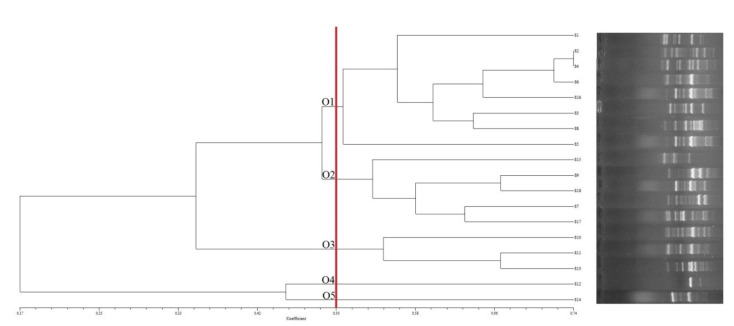
UPGMA dendrogram of the *C. perfringens* isolates from the raw meat samples based on the OPA-3 PCR analysis.

**Table 1 antibiotics-11-00340-t001:** Antibiotic resistance phenotype of *C. perfringens* isolated from the raw beef meat samples.

Antibiotic Class	Antibiotic Agent	n (%)		
		Whole Meat (n = 10)	Minced Meat (n = 8)	Total (n = 18)
β-Lactams	imipenem	4 (40.0)	1 (12.5)	5 (27.7)
	amoxicillin	7 (70.0)	4 (50.0)	11 (61.1)
	ampicillin	7 (70.0)	6 (75.0)	13 (72.2)
	cefepime	3 (30.0)	0 (0.0)	3 (16.6)
Cephalosporins	ceftriaxone	4 (40.0)	0 (0.0)	4 (22.2)
Aminoglycosides	amikacin	2 (20.0)	1 (12.5)	3 (16.6)
Fluoroquinolones	ciprofloxacin	5 (50.0)	2 (25.0)	7 (38.8)
Phenicols	chloramphenicol	3 (30.0)	3 (37.5)	6 (33.3)
Tetracyclines	tetracycline	7 (70.0)	5 (62.5)	12 (66.6)

**Table 2 antibiotics-11-00340-t002:** Patterns of multidrug resistance classes of the *C. perfringens* isolates from the raw meat samples.

No. Classes of Antibiotics	Patterns of Multidrug Resistance ^a^ (No. Isolates in Each Pattern)	No. Total Isolates (%) (n = 18)
One	βLs (n = 1)	3 (16.6)
	TCs (n = 2)	
Two	βLs-TCs (n = 5)	8 (44.4)
	βLs-CPs (n = 1)	
	βLs-PNs (n = 1)	
	QNs-TCs (n = 1)	
Three	βLs-TCs-QNs (n = 1)	3 (16.6)
	βLs-PNs-QNs (n = 1)	
	βLs-TCs-CPs (n = 1)	
Four	βLs-PNs-QNs-AGs (n = 1)	2 (11.1)
	βLs-PNs-QNs-TCs (n = 1)	
Five	βLs-CPs-QNs-TCs-AGs (n = 1)	1 (5.5)
Six	βLs-CPs-QNs-TCs-AGs-PNs (n = 1)	1 (5.5)

^a^ βLs, β-Lactams; AGs, Aminoglycosides; TCs, Tetracyclines; PNs, Phenicols; QNs, Fluoroquinolones; CPs, Cephalosporins.

**Table 3 antibiotics-11-00340-t003:** Resistance phenotype, toxin genes, toxinotypes and OPA-3 genotypes in *C. perfringens* isolated from the raw whole and minced meat samples.

No. Sample	Isolate	Source	Resistance Phenotype ^a^	Toxin Genes	Toxin Type	OPA-3 Group
1	CPQM19-1	Whole meat	TET and CIP	*cpa*+	A	O1
2	CPQM19-2	Whole meat	IPM, AMX, AMK, AMP, TET, FEP, CHL, CIP, and CRO	*cpa*+	A	O1
3	CPQM19-3	Whole meat	AMX, TET, and AMP	*cpa*+ *cpe*+	Ae	O1
4	CPQM19-4	Whole meat	CIP, AMX, AMP, and TET	*cpa*+	A	O1
5	CPQM19-5	Whole meat	CRO, FEP, AMX, AMP, and TET	*cpa*+ *etx*+	D	O1
6	CPQM19-6	Whole meat	CRO and AMP	*cpa*+	A	O1
7	CPQM19-7	Whole meat	FEP and CHL	*cpa*+	A	O2
8	CPQM19-8	Minced meat	TET and AMP	*cpa*+ *etx*+	D	O1
9	CPQM19-9	Minced meat	AMK, CIP, AMX, AMP, and CHL	*cpa*+ *cpe*+	Ae	O2
10	CPQM19-10	Minced meat	AMP	*cpa*+ *cpe*+ *cpb*+	Ce	O3
11	CPQM19-11	Minced meat	TET	*cpa*+ *cpb*+ *etx*+	B	O3
12	CPQM19-12	Minced meat	IPM, CIP, AMX, AMP, TET, and CHL	*cpa*+ *etx*+	D	O4
13	CPQM19-13	Minced meat	TET	*cpa*+ *cpe*+	Ae	O3
14	CPQM19-14	Minced meat	AMX, AMP, and TET	*cpa*+ *iap*+	E	O5
15	CPQM19-15	Whole meat	IPM, AMX, AMP, and TET	*cpa*+ *etx*+	D	O2
16	CPQM19-16	Whole meat	IPM, AMX, AMK, AMP, TET, CIP, and CRO	*cpa*+	A	O1
17	CPQM19-17	Whole meat	IPM, CIP, AMX, and CHL	*cpa*+	A	O2
18	CPQM19-18	Minced meat	FEP, AMX, and TET	*cpa*+ *cpe*+	Ae	O2

^a^ IPM, imipenem; AMX, amoxicillin; AMK, amikacin; AMP, ampicillin; TET, tetracycline; FEP, cefepime; CHL, chloramphenicol; CIP, ciprofloxacin; CRO, ceftriaxone.

**Table 4 antibiotics-11-00340-t004:** Primer sequences used in this study for the genotyping and detection of *C. perfringens* toxin-encoding genes.

Primer	Sequence (5′-3′)	Annealing Temperature (°C)	Amplicon (bp)
*Cpa*	AGTCTACGCTTGGGATGGAA	56	900
	TTTCCTGGGTTGTCCATTTC		
*Cpe*	GGGGAACCCTCAGTAGTTTCA	56	506
	ACCAGCTGGATTTGAGTTTAATG		
*Cpb*	TCCTTTCTTGAGGGAGGATAAA	56	611
	TGAACCTCCTATTTTGTATCCCA		
*Etx*	TGGGAACTTCGATACAAGCA	56	396
	TTAACTCATCTCCCATAACTGCAC		
*Iap*	AAACGCATTAAAGCTCACACC	56	293
	CTGCATAACCTGGAATGGCT		
OPA-3	AGTCAGCCAC	42	−

## Data Availability

We confirm that all data included in this study are available within the article.

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
