# Peer review of "Prevalence, Antibiotic Resistance, Toxin-Typing and Genotyping of *Clostridium perfringens* in Raw Beef Meats Obtained from Qazvin City, Iran"

_antibiotics, 2022, doi:10.3390/antibiotics11030340_

Round 1
Reviewer 1 Report
The design of the isolation of the pathogenic agent seems adequate to me, as well as the techniques used. However, I do not find anything new in the investigation. The resistance of antibiotics to pathogenic microorganisms is known to be the product of irresponsible bad management of the use of drugs in domestic animals. Diseases transmitted by food, whether in raw meat, miced meat or whole meat when handled under conditions of poor sanitary practices, will have a higher prevalence to transmit diseases due to the consumption of contaminated food. In this case, what would be the novelty of this research? In the discussion it is necessary to highlight more about it, so that it is clear.
Author Response
Dear Reviewer 1
There are limited studies investigating antibiotic resistance and toxin types in C. perfringens, known as gram-positive anaerobic foodborne bacterial pathogens, isolated from food samples. In this study, we performed research to investigate antimicrobial susceptibility, toxin encoding genes and genotypic relationships among these properties in C. perfringens isolated from raw meat samples as the main foods contaminated with this foodborne pathogen.
Reviewer 2 Report
This is a manuscript review for the article 'Prevalence, antibiotic resistance, toxin-typing and genotyping of Clostridium perfringens in raw beef meats'
The article is well written and brings some new insights. However, the following issues below will need addressing.
TITLE and ABSTRACT
The title should be changed to 'Prevalence, antibiotic resistance, toxin-typing and genotyping of Clostridium perfringens in raw beef meats obtained from Qazvin city, Iran.' This is because the sample size is small and the study will help geographical comparisons.
The location where beef was sourced should also be reflected in the abstract.
Line 17. It is not conventional to start sentences with figures. Re-write.
Up to 44.4 - 55.5% of these.................
Line 20: 38.8% of the isolates were multi-drug resistant. Change to 'Multi-drug resistance was found in 38% of the isolates.'
Line 36: ..........C. perfringens, known as the second most prevalent bacterial pathogens........ Provide a reference for this statement.
RESULTS AND DISCUSSION
Information in Tables 3 and 4 are repeated in Table 5. This is a duplication and should be avoided. Remove Tables 3 and 4.
Line 287: 'Our findings are corresponding' Change to 'Our findings corresponds'
Line 288: Provide the references for the countries mentioned.
Line 313: Change 'Chukwu et al. (2017)' to' Chukwu et al. [19]
Line 314: 'change Llanco et al. (2015)' to 'Llanco et al. [46]'
Check the rest of the document and make sure that the correct in-text citation format is followed.
Klebsiella, Ecoli and Staphylococcus were used as control but the results were not shown or mentioned. This should be addressed.
Apart from the significant values, the statistics test Chi-square and Fisher's tests mentioned in the method section were not specifically mentioned in the results or discussed. This should be addressed.
REFERENCES
This has not been written as per the guide. Authors should consult the journal's guide
Author Response
Dear Reviewer 2
Title and Abstract
- The title has been revised according to your comments.
- The location where beef was sourced is added into the text in the abstract section.
- Line 17: the sentence is revised.
- Line 20: the sentence is revised according to your comment.
- Line 36: the sentence is revised and the reference is added to the text.
Results and Discussion
- Tables 3 and 4 are removed and other sections are revised according to this change throughout the manuscript.
- Line 287: the sentence is revised according to your comment.
- Line 288: the sentence is revised according to your comment.
- Line 313: the sentence is revised according to your comment.
- Line 314: the sentence is revised according to your comment.
- The rest of the document is checked and revised throughout the manuscript in the correct in-text citation format.
- Klebsiella, E. coli and Staphylococcus were used just as the negative and positive controls in AMR evaluation methods. This statement is added to the text.
- Chi-square significance levels are addressed throughout the manuscript.
References
- All references are revised according to the journal standards.
Reviewer 3 Report
The purpose of this study was to determine the prevalence rate, antimicrobial susceptibility, toxinotype, and genetic diversity in C. perfringens isolates from raw whole and minced meat samples. The manuscript is well written and there are few comments that will need to be addressed:
1- The number of the collected samples is low
2- Please provide the statistical analysis in a separate section.
3- What do you mean by all experiments were repeated 3 times?
4- Remove the results and % that you provided in the conclusion section
5- Please provide correlation analysis between phenotypic and genotypic resistance as provided in these papers (for example), you can also cite them and other papers if you want when you add this section to the methodology part
a- https://www.mdpi.com/2079-6382/10/12/1450
b- https://www.nature.com/articles/s41598-018-23962-7
6- In table 2: what is the difference of samples number in column 2 and in column 3?
7- Please remove the repeated results as much as you can from the discussion section.
8- The discussion is very long, please shorten it up. you do not have to keep comparing your results with all the other results in the world, just select the significant one, which may be in the same study area or country.
Author Response
Dear Reviewer 3
- The main limitation in this study is the relatively small sample size which is mentioned and added into the discussion and conclusion sections of the manuscript.
- Statistical analysis is provided in a separate section.
- All measurements were repeated 3 times. This section is revised in the text.
- All results and % provided in the conclusion section are removed from the text.
- Correlation analysis also can be provided through the investigation of the genotypic relatedness among the isolates and describes the relationship among the phenotypic and toxin type profiles of the isolates which are implemented in this study. This relationship has previously been demonstrated via genotyping analysis (Chon JW, Seo KH, Bae D, Park JH, Khan S, Sung K. Prevalence, toxin gene profile, antibiotic resistance, and molecular characterization of Clostridium perfringens from diarrheic and non-diarrheic dogs in Korea. Journal of veterinary science. 2018 May 1;19(3):368-74.).
- In Table 2: Column 1 provided different patterns of multi-drug resistance in each category of antibiotic classes and column 2 demonstrated the number of isolates in each category of antibiotic classes.
- All repeated results are removed and revised throughout the discussion section.
- The discussion section is revised to a shortened form and some unnecessary texts are removed from this section of the manuscript.
Round 2
Reviewer 3 Report
The authors addressed all my comments
Thank you
Author Response
Dear Editor
The manuscript entitled “Prevalence, antibiotic resistance, toxin-typing and genotyping of Clostridium perfringens in raw beef meats” is revised according to the reviewers` comments. The cuts chosen in this study as raw whole and minced meat samples were taken from Shank and Tenderloin parts of the slaughtered beef cattle. This statement is added to the materials and methods section, sample collection part.
Dear Reviewer 1
There are limited studies investigated the antibiotic resistance and toxinotypes in C. perfringens, as a gram-positive anaerobic foodborne bacterial pathogen, isolated from food samples. In this study we preformed research to investigate antimicrobial susceptibility, toxin encoding genes and genotypic relationship among these properties in C. perfringens isolated from raw meat samples as the main foods contaminated with this foodborne pathogen.
Dear Reviewer 2
Title and Abstract
- Title is revised according to your comment.
- The location where beef was sourced is added into the text in abstract section.
- Line 17: the sentence is revised.
- Line 20: the sentence is revised according to your comment.
- Line 36: the sentence is revised and the reference is added into the text.
Results and Discussion
- Tables 3 and 4 are removed and other sections are revised according to this change throughout the manuscript.
- Line 287: the sentence is revised according to your comment.
- Line 288: the sentence is revised according to your comment.
- Line 313: the sentence is revised according to your comment.
- Line 314: the sentence is revised according to your comment.
- Rest of the document are checked and revised throughout the manuscript for correct in-text citation format.
- Klebsiella, E. coli and Staphylococcus were used just as the negative and positive controls in AMR evaluation methods. This statement is added into the text.
- Chi-square significance levels are addressed throughout the manuscript.
References
- All references are revised according to the journal standards.
Dear Reviewer 3
- The main limitation in this study is relatively small sample size which is mentioned and added into the discussion and conclusion sections of the manuscript.
- Statistical analysis is provided in a separate section.
- All measurements were repeated 3 times. This section is revised in the text.
- All results and % provided in the conclusion section are removed from the text.
- Correlation analysis also can be provided through the investigation of the genotypic relatedness among the isolates and description the relationship among the phenotypic and toxinotype profiles of the isolates which is implemented in this study. This relationship has previously been demonstrated via genotyping analysis (Chon JW, Seo KH, Bae D, Park JH, Khan S, Sung K. Prevalence, toxin gene profile, antibiotic resistance, and molecular characterization of Clostridium perfringens from diarrheic and non-diarrheic dogs in Korea. Journal of veterinary science. 2018 May 1;19(3):368-74.).
- In Table 2: Column 1 provided different patterns of multi-drug resistance in each category of antibiotic classes and column 2 demonstrated the number of the isolates in each category of antibiotic classes.
- All repeated results are removed and revised throughout the discussion section.
- Discussion section is revised to a shortened form and some unnecessary texts are removed from this section in the manuscript.
Kind regards
